# The CAPRA&PDE4D5/7/9 Prognostic Model Is Significantly Associated with Adverse Post-Surgical Pathology Outcomes

**DOI:** 10.3390/cancers15010262

**Published:** 2022-12-30

**Authors:** Chloe Gulliver, Sebastian Huss, Axel Semjonow, George S. Baillie, Ralf Hoffmann

**Affiliations:** 1School of Cardiovascular and Metabolic Health, College of Medical, Veterinary and Life Science, University of Glasgow, Glasgow, G12 8QQ, UK; 2Gerhard-Domagk-Institute of Pathology, University Hospital Münster, 48149 Münster, Germany; 3Prostate Center, University Hospital Münster, 48149 Münster, Germany; 4Oncology Solutions, Philips Research Europe, 5656AE Eindhoven, The Netherlands

**Keywords:** phosphodiesterase, prostate cancer, risk stratification, prognosis, active surveillance, molecular biomarker

## Abstract

**Simple Summary:**

PDE4D5, PDE4D7 and PDE4D9 are prostate expressed transcripts of the PDE4D gene coding for cAMP degrading phosphodiesterases. These genes have been implicated in the change of prostate cancer from an androgen sensitive to an androgen insensitive, treatment resistant state. CAPRA is a clinical risk model built from patient demographic data (e.g., age) and clinical variables (e.g., PSA, biopsy Gleason score). The gene expression of the PDE4D transcripts is measured on the extracted RNA from a patient’s tumor sample. We have previously published that the clinical-genomic risk score CAPRA&PDE4D5/7/9, which is a combination model of the CAPRA score with the expression levels of the respective PDE4D transcripts, is associated with prostate cancer progression after surgical removal of the prostate. Here we show that this risk score is also associated with adverse pathology features like an elevated Gleason score or extended tumor growth into or beyond the prostate capsule or into the pelvic lymph nodes. For this we determined the CAPRA&PDE4D5/7/9 risk score in a cohort of patients who all underwent systematic needle biopsy followed by radical prostatectomy as a primary treatment. We determined the negative predictive value (NPV) for CAPRA&PDE4D5/7/9 risk score in a low-to-intermediate sub-cohort by applying a pre-defined cut-off. This selected low CAPRA&PDE4D5/7/9 risk cohort demonstrated high NPV for negative adverse pathology and might therefore represent a suitable patient group to be managed by active surveillance.

**Abstract:**

Objectives: To investigate the association of the prognostic risk score CAPRA&PDE4D5/7/9 as measured on pre-surgical diagnostic needle biopsy tissue with pathological outcomes after radical prostatectomies in a clinically low–intermediate-risk patient cohort. Patients and Methods: RNA was extracted from biopsy punches of diagnostic needle biopsies. The patient cohort comprises *n* = 151 patients; of those *n* = 84 had low–intermediate clinical risk based on the CAPRA score and DRE clinical stage <cT3. This cohort (*n* = 84) was investigated for pathology outcomes in this study. RT-qPCR was performed to determine PDE4D5, PDE4D7 and PDE4D9 transcript scores in the cohorts. The CAPRA score was inferred from the relevant clinical data (patient age, PSA, cT, biopsy Gleason, and percentage tumor positive biopsy cores). Logistic regression was used to combine the PDE4D5, PDE4D7 and PDE4D9 scores to build a PDE4D5/7/9_BCR regression model. The CAPRA&PDE4D5/7/9_BCR risk score used was same as previously published. Results: We investigated three post-surgical outcomes in this study: (i) Adverse Pathology (any ISUP pathological Gleason grade >2, or pathological pT stage > pT3a, or tumor penetrated prostate capsular status, or pN1 disease); (ii) any ISUP pathological Gleason >2; (iii) any ISUP pathological Gleason >1. In the *n* = 84 patients with low to intermediate clinical risk profiles, the clinical-genomics CAPRA&PDE4D5/7/9_BCR risk score was significantly lower in patients with favorable vs. unfavorable outcomes. In univariable logistic regression modeling the genomics PDE4D5/7/9_BCR as well as the clinical-genomics CAPRA&PDE4D5/7/9_BCR combination model were significantly associated with all three post-surgical pathology outcomes (*p* = 0.02, *p* = 0.0004, *p* = 0.04; and *p* = 0.01, *p* = 0.0002, *p* = 0.01, respectively). The clinically used PRIAS criteria for the selection of low-risk candidate patients for active surveillance (AS) were not significantly associated with any of the three tested post-operative pathology outcomes (*p* = 0.3, *p* = 0.1, *p* = 0.1, respectively). In multivariable analysis adjusted for the CAPRA score, the genomics PDE4D5/7/9_BCR risk score remained significant for the outcomes of adverse pathology (*p* = 0.04) and ISUP pathological Gleason >2 (*p* = 0.004). The negative predictive value of the CAPRA&PDE4D5/7/9_BCR risk score using the low-risk cut-off (0.1) for the three pathological endpoints was 82.0%, 100%, and 59.1%, respectively for a selected low-risk cohort of *n* = 22 patients (26.2% of the entire cohort) compared to 72.1%, 94.4%, and 55.6% for *n* = 18 low-risk patients (21.4% of the total cohort) selected based on the PRIAS inclusion criteria. **Conclusion:** In this study, we have shown that the previously reported clinical-genomics prostate cancer risk model CAPRA&PDE4D5/7/9_BCR which was developed to predict biological outcomes after surgery of primary prostate cancer is also significantly associated with post-surgical pathology outcomes. The risk score predicts adverse pathology independent of the clinical risk metrics. Compared to clinically used active surveillance inclusion criteria, the clinical-genomics CAPRA&PDE4D5/7/9_BCR risk model selects 22% (*n* = 8) more low-risk patients with higher negative predictive value to experience unfavorable post-operative pathology outcomes.

## 1. Introduction

Prostate cancer has developed into the second most cancer site in men worldwide with an estimated 1.4 million newly diagnosed cases in 2020 [1]. Age-standardized incidence rates vary by around >10-fold with highest rates per 100,000 men observed in Northern and Western Europe (83 and 78, respectively) followed by Caribbean, Australia/New Zealand (76) and Northern America (73) while much lower incidence rates are seen in Africa (17–41) and Asia (<15). Mortality due to prostate cancer accounts for 6.8% (375,000 cases annually) of all cancer death in men [1].

Survival in prostate cancer is largely related to the diagnosis of low-grade disease on final diagnosis after primary treatment. The most powerful measure to assess the individual risk of prostate cancer progression or death of disease is the Gleason scoring system which was first introduced in 1966 [2] and subsequently modified in 2005, and again in 2014 [3,4]. Various types of studies including non-randomized, as well as randomized trials, have demonstrated that outcomes for patients with pathological Gleason ≤6 (ISUP grade 1) tumors have similar outcomes irrespective of whether and how they were treated. It is generally accepted that these patients have an excellent 10–15-year survival probability with minimal risk of disease specific death [5,6,7]. However, compared to endpoints after post-surgical pathology, the biopsy assessed Gleason score underestimates the final grade as well as the extent of the disease due the so-called sampling error. In a recent large cohort of more than 7000 patients undergoing radical prostatectomy after systematic ≥10-core needle biopsy 36.3% of biopsy Gleason ≤6 was upgraded to ≥6 after post-surgical pathology. Of the biopsies with a Gleason score of 3 + 4 (ISUP grade 2) around 50% were assessed with a matching Gleason score of 3+4. Around 25% of biopsies were downgraded to Gleason score <3 + 4 (ISUP grade 1) and the other 25% biopsies were upgraded to a Gleason score >3 + 4 (ISUP grades 3–5) [8]. Moreover, in addition to grade migration from biopsy to post-surgery pathology, patients may harbor other adverse features indicting more extensive, non-organ confined disease. This has been illustrated in a study on patients with biopsy Gleason ≤6 who were eligible for active surveillance (AS) based on various including metrics. Of those men, 20–30%, depending on the AS inclusion criteria used, were low-grade and organ-confined after surgery and pathology assessment. Up to 10% of these patients were identified with extra-prostatic extension (EPE) and for up to 50% positive surgical resection margins were observed in pathology review [9]. Thus, more accurate prediction of pathological outcomes after surgery is required for optimal treatment decision making of patients with primary prostate cancer.

Androgens are the key drivers of prostate cancer growth and progression. The androgen receptor (AR) transduces intracellular signaling following binding of androgen hormones such as dihydrotestosterone (DHT) in the cytoplasm of the cell. Within the cytoplasm, AR is bound to HSP90 which maintains the nuclear receptor in a high-affinity confirmation for ligand binding [10]. Upon DHT binding, the AR dissociates from HSP90 which enables nuclear translocation and start of the androgen related transcriptional program.

Cross-talk between the AR pathway and the cyclic-AMP (cAMP)/protein kinase A (PKA) pathway has been observed previously [11]. The interaction between these important signaling axes is supported by evidence from clinical data where the over-expression of the specific forms of the catalytic [12,13] and regulatory subunits of PKA have demonstrated association with disease progression and poor patient outcome [14]. Additionally, confirmatory in vitro data were collected following the stimulation of AR transactivation and enhanced downstream PSA transcription by elevation of cAMP and subsequent activation of PKA [15]. However, only recently data were presented for outlining a molecular mechanism explaining how cAMP/PKA activation may mediate stimulation of AR signaling. In this model, PKA activity is essential for AR nuclear translocation by phosphorylation of HSP90, thereby releasing AR from its complex with this heat-shock protein to enable AR nuclear migration via binding and co-transport with HSP27 [16]. Thus, cAMP/PKA signaling and activity is identified as a prerequisite for classical prostate AR signaling.

The catalytic activity of phosphodiesterases (PDEs) provides the sole means to degrade the important second messenger 3′-5′-cAMP and hence have the unique ability to regulate the spatial and temporal dynamics of cAMP signaling. Eleven gene families with multiple members and various transcripts per family member have been described and are extensively reviewed elsewhere [17,18]. We previously identified the long PDE4D isoform PDE4D7 as a key player in the development and progression of prostate cancer. High levels of PDE4D7 expression is associated with androgen sensitivity of prostate cancer cells while diminished PDE4D7 transcription in prostate cancer is strongly correlated to androgen resistance [19,20]. Elevated expression of PDE4D7 is associated with the presence of the TMPRSS2-ERG fusion in prostate cancer cell lines and human tumor tissue [21]. Moreover, PDE4D7 expression is inversely correlated with adverse biological outcomes such as PSA relapse after radical prostatectomy [22,23] which was further modeled in combination with other long PDE4D isoforms (i.e., PDE4D5 and PDE4D9) adjusted for the clinical prognostic CAPRA score [24]. Overall, we have described PDE4D7 as novel biomarker to support the classification of prostate cancer into those with very low risk of disease progression (in the case of high PDE4D7 expression) compared to those with elevated risk of post-treatment disease progression (in the case of low PDE4D7 expression) [25].

Here, we set out to investigate the association of the previously published prognostic combination model of PDE4D5, PDE4D7, PDE4D9 with the CAPRA score (CAPRA&PDE4D5/7/9 model [24]) to adverse pathological outcomes after radical prostatectomy in a clinically low-to-intermediate-risk patient cohort compared to the CAPRA score, to a PDE4D5, PDE4D7, PDE4D9 combination model, as well as to the two selected AS inclusion models PRIAS and UCSF [9].

## 2. Patients and Methods

### 2.1. Patient Cohort and Samples

Patients were diagnosed at a single clinical center in Germany and were undergoing radical prostatectomy (RP) between 1994–2011. The aggregated characteristics of the total patient cohort as well as of the sub-cohort of low–intermediate clinical risk based on CAPRA score ≤5 and clinical cT <3 and complete data on PDE4D5, PDE5D7, and PDE4D9 expression (*n* = 84) are summarized in Table 1. From the tumor positive pre-surgical diagnostic needle biopsy with the highest Gleason grade a single biopsy punch (~1 × 2 mm) was collected per patient for RNA extraction and down-stream molecular biology analysis. 

### 2.2. Clinical Risk Metrics

The CAPRA score was developed to predict the risk of post-surgical disease progression based on pre-surgical clinical variables pre-operative PSA, biopsy Gleason score, clinical stage, percentage of tumor positive biopsies, and patient age at diagnosis [26]. In short, the risk of post-surgical progression is represented by an absolute score on a 0–10 scale with three clinical risk categories: low risk: CAPRA scores 0–2; intermediate risk: CAPRA scores 3–5; high risk: CAPRA scores 6–10. PRIAS is one of the rule-based metrics for inclusion of patients with primary prostate cancer into active surveillance (AS). Patients are eligible for AS inclusion with pre-operative PSA <10 ng/mL, biopsy ISUP Gleason grade 1 (Gleason score 3 + 3), and PSA density <0.2 ng/mL PSA/mL prostate volume, a clinical stage cT ≤ cT2, and number of tumor positive biopsy cores ≤2 which is similar as compared to other AS inclusion metrics [9]. We selected PRIAS as a representative metric to select patients eligible for AS because the input clinical data were all available for our patient cohort.

### 2.3. Laboratory Methods

All molecular laboratory methods including oligonucleotide primers and probes for RT-qPCR (reverse transcriptase quantitative PCR), RNA extraction, as well as data quality control and procedures to include/exclude measurements from the statistical analysis, were used as previously described by us [22,23,24].

### 2.4. Data Analysis and Statistics

Calculation of the normalized PDE4D transcript expression (i.e., for PDE4D5, PDE4D7, and PDE4D9) was performed by subtracting the RT-qPCR Cq of the respective PDE4D transcript from the averaged RT-qPCR Cq of four selected reference genes [23]. The normalized expression values for the PDE4D5, PDE4D7, and PDE4D9 genes were used as inputs into the previously published logistic combination regression model of the CAPRA score with the PDE4D5, PDE4D7, and PDE4D9 genes (the CAPRA&PDE4D5/7/9 model score = −3.1 + (−0.76 × PDE4D5_norm) + (−0.7 × PDE4D7_norm) + (−0.73 × PDE4D9_norm) + (0.65 × CAPRA Score) [24]). Furthermore, we created a new logistic regression model on the entire patient cohort (*n* = 151) using post-surgical BCR (biochemical relapse) as an endpoint, only using the normalized expression values for the PDE4D transcripts (i.e., PDE4D5, PDE4D7, and PDE4D9) without adding any clinical parameters into the model to create the PDE4D5/7/9_BCR model = −1.11 + (−0.47 × PDE4D5_norm) + (−0.42 × PDE4D7_norm) + (−0.78 × PDE4D9_norm).

Uni- and multi-variate logistic regression analyses were applied to correlate the CAPRA scores, the PDE4D5/7/9 scores, and the CAPRA&PDE4D5/7/9 scores, and the PRIAS metric to adverse pathology outcomes after initial surgery in the clinically low–intermediate-risk cohort (*n* = 84). For statistical analysis the software package MedCalc (MedCalc Software BVBA, Ostend, Belgium) was used. *p*-values < 0.05 were regarded statistically significant.

## 3. Results

### 3.1. Patient Demographics

Patients (*n* = 151) were selected from a single treatment center in Germany based on the availability of relevant clinical and outcome data and access to patient material from diagnostic needle biopsies. All patients were treated by radical prostatectomy and post-surgical pathology was available. The post-surgical median follow-up for this patient cohort was 73.6 months (Table 1). In this patient group 25.2% were classified as clinically low risk, 54.3% as clinically intermediate risk, and 20.5% as clinically high risk based on the CAPRA score categories (CAPRA scores 0–2: low risk; CAPRA scores 3–5: intermediate risk; CAPRA scores >5: high risk) (Table 1).

Of the total patient cohort, we selected a low–intermediate-risk cohort (Table 1) based on the conceptual idea that such patient group would be more eligible for testing of low-risk candidates for inclusion to active surveillance regimes. Selection was made based on the following inclusion criteria: CAPRA score ≤5 and DRE clinical stage ≤cT2. In this selected group (*n* = 84), 38.1% (*n* = 35) had clinically low-risk disease (CAPRA score 0–2), 61.9% (*n* = 57) had clinically intermediate-risk disease (CARPA score 3–5), while no high-risk patients remained in the cohort based on the CAPRA score categories. The median post-surgical follow-up of these selected patients was 82.1 months. We used this low–intermediate clinical risk sub-cohort for uni- and multi-variable regression analysis. Finally, we compared the negative predictive value (NPV) and positive predictive value (PPV) for this patient sub-group with those of the entire patient cohort.

### 3.2. Kaplan–Meier Survival Analysis of the CAPRA&PDE4D5/7/9_BCR Logistic Regression Model

In the previous setting we transformed the logit(p) values as derived from the CAPRA&PDE4D5/7/9 logistic regression model to a 1–5 score distribution and categorized patients into four different risk groups to experience post-surgical biochemical relapse (BCR) [24]. Here, we calculated the probability (p) from the logit function of the regression model for each patient which provides an individual risk to experience post-surgical BCR on a scale from 0–1 (or 0–100%). We selected two cut-offs (0.1 and 0.835) which stratifies patients into three different risk groups (instead of previously four risk groups) to experience BCR after radical prostatectomy (RP) (low risk [*p* < 0.1], intermediate risk [0.1 < *p* ≤ 0,835], high risk [*p* > 0.835]). The cut-offs were selected such that the low-risk group (*p* < 0.1; *n* = 29) represents the previously published CARPA&PDE4D5/7/9 risk group with scores 1–2 (*n* = 29; [24]); likewise, the intermediate-risk group (*n* = 80) defined by the probability (p) represents the CARPA&PDE4D5/7/9 risk group with scores 2–3 (*n* = 80; [24]); and the high-risk group (*n* = 42) as defined here represents the previously presented CARPA&PDE4D5/7/9 risk group with scores 3–5 (*n* = 42; [24]). The risk scores of the here presented CARPA&PDE4D5/7/9_BCR regression model represent the prognostic risk to predict post-surgical patient outcomes (Figure 1).

### 3.3. Analysis of CAPRA&PDE4D5/7/9_BCR Risk Model

We first tested whether the mean risk score of the CAPRA&PDE4D5/7/9_BCR model was statistically significantly different between various outcome groups of the selected low–intermediate patient cohort (*n* = 84). Three study outcomes of adverse pathology (AP) after prostate resection were defined as: (i) any ISUP pathological Gleason grade >2, or pathological pT stage > pT3a, or tumor penetrated prostate capsular status, or pN1 disease; (ii) any ISUP pathological Gleason >2; (iii) any ISUP pathological Gleason >1. For all outcomes we observed a significant difference in the mean CAPRA&PDE4D5/7/9_BCR risk score between the patient groups comparing favorable vs. unfavorable outcomes (*p* = 0.006, *p* < 0.0001, *p* = 0.002, respectively; Table 2).

### 3.4. Univariable Logistic Regression (UVLR) Analysis

Next, we examined the CAPRA risk score (the base model), the PDE4D5/7/9_BCR (the PDE transcript model), as well as the CAPRA&PDE4D5/7/9_BCR (the clinical-PDE transcript combination model) scores in univariable logistic regression analysis for their association with the adverse RP outcomes as outlined above. 

The CAPRA score alone was not statistically significantly associated with AP in UVLR (OR = 1.4; *p* = 0.09). In contrast, both the PDE4D5/7/9_BCR model as well as the CAPRA&PDE4D5/7/9_BCR model were significantly associated with adverse pathology after surgery (OR = 1.6; *p* = 0.02 and OR = 1.4; *p* = 0.01, respectively; Table 3). The Area’s under the ROC Curves (AUROC’s) to correctly diagnose the endpoint of AP were 0.6, 0.67, and 0.68 for the CAPRA score, the PDE4D5/7/9_BCR model, and the CAPRA&PDE4D5/7/9_BCR model, respectively.

The CAPRA score was significantly associated with the endpoint of pathology ISUP Gleason grade >2 or >1 (OR = 2.4, *p* = 0.002, and OR = 1.6, *p* = 0.02, respectively) when testing the association in logistic regression modeling. The PDE transcript model and the clinical-PDE transcript combination model demonstrated higher AUROC for the RP outcome ISUP Gleason grade >2 compared to the CAPRA score (AUROC = 082, OR = 2.5; *p* = 0.0004, and AUROC = 0.74, OR = 1.9, *p* = 0.0002, respectively), while for the endpoint ISUP Gleason grade > 1 the significance of the association of the three tested models were comparable (Table 3). However, the CAPRA&PDE4D5/7/9_BCR model showed an increase in AUROC to correctly assess the three tested adverse post-interventional pathology outcomes compared to the base model of the CARPA score alone by 8, 8, and 3 units, respectively (Table 3). This represents a substantial increase for the post-surgery endpoints AP and ISUP Gleason grade > 2 over the sole use of the clinical metric CAPRA alone (Table 3).

Further, we compared how the clinically used active surveillance (AS) inclusion metric PRIAS predicted post-surgical adverse pathology outcomes. In total, *n* = 18 out of 84 (21.4%) patients were defined as eligible for AS by the PRIAS criteria in the selected low–intermediate risk sub-cohort.

These data demonstrate that the previously reported combination risk model of the normalized expression of PDE4D5, PDE4D7, and PDE4D9 together with the CAPRA score is significantly associated with adverse outcomes at post-surgical pathology.

### 3.5. Multivariable Logistic Regression (MVLR) Modeling

In the MVLR analysis we set out to test the CAPRA model and the PDE4D5/7/9_BCR model for independent association with the three post-surgical pathology endpoints. We observed a significant association of the PDE4D5/7/9_BCR model for Adverse Pathology (AP) as endpoint but not for the CAPRA score (OR = 1.5, *p* = 0.04; OR = 1.3, *p* = 0.3, respectively; Table 4). The AUROC of the combined MVLR model was calculated as 0.67 which represents the AUROC (0.68) of the CAPRA&PDE4D5/7/9_BCR model as tested in UVLR (Table 3). Concerning the two endpoints of pathology ISUP Gleason grade >2 or >1, the two tested models were significantly associated with ISUP Gleason grade >2 but not with ISUP Gleason grade >1 (Table 4) which may indicate that both variables contribute equally to the MVLR model given that the combination model CAPRA&PDE4D5/7/9_BCR was significantly associated with the ISUP Gleason grade >1 endpoint in UVLR (Table 3). 

Taken together, these results provide evidence that the logistic regression model PDE4D5/7/9_BCR is significantly and independently associated with adverse RP outcomes in multi-variate modelling with the CAPRA score. Adding the PDE4D transcript expression to the CAPRA score adds value to the molecular model for the prediction of adverse post-surgical pathology outcomes.

### 3.6. Negative Predictive Values

Next, we investigated the negative predictive value (NPV) by applying a cut-off of 0.1 to the CAPRA&PDE4D5/7/9_BCR score to define a low-risk group vs. an intermediate–high-risk group (≥0.1). Using this cut-off, *n* = 22 out of 84 patients (26.1%) were selected as low risk according to the CAPRA&PDE4D5/7/9_BCR risk model. We calculated the NPV (TN/(TN + FN)) for various post-surgical endpoints (Table 5). These endpoints were either pathology outcome based (pathology IUSP Gleason grade, pathology pT, capsular status, surgical margin status, lymph node invasion status) or related to longitudinal outcomes (post-surgical BCR, post-surgical start of secondary therapies). The NPV was determined to be 100% for pathology IUSP Gleason grade ≤2 outcome after operation, any pT stage > ≤pT3a, freedom of lymph node invasion, freedom of post-operative BCR and start of any secondary treatment due to disease progression. The NPV’s for some of the other tested adverse pathology outcomes were around 80% (e.g., freedom of tumor penetrated capsular status, positive surgical margins, or any pathology stage ≤T2) while being ~60% for pathology IUSP Gleason grade ≤1. However, although some of these outcomes are risk factors for disease progression after primary intervention, none of the affected patients showed any sign of progressive disease during the >7.5 years of follow-up after RP. The clinical characteristics of the *n* = 22 CAPRA&PDE4D5/7/9_BCR low-risk patients are summarized in Table 1.

The NPV’s for the adverse pathology outcomes for the PRIAS inclusion low-risk patient group [*n* = 18 out of 84 (21.4%)] were generally 5–10% lower compared to the NPV’s of the CAPRA&PDE4D5/7/9_BCR low-risk inclusion cut-off (Table 5). It is notable that the overlap of patients selected by both models is only *n* = 10 subjects, indicating that the definition of low-risk patients is to a significant extent perpendicular to the low-risk stratification by the PRIAS AS metrics.

Finally, we calculated the NPV’s of the CAPRA&PDE4D5/7/9_ BCR low-risk inclusion cut-off for the entire patient cohort (*n* = 151) compared to the PRIAS inclusion criteria. It is evident that the NPV’s to diagnose adverse pathology for both patient groups are very similar. However, using the CAPRA&PDE4D5/7/9_ BCR model selects *n* = 11 (61.1%) more patients to be low risk compared to PRIAS (Table 6). The PPV’s for adverse outcomes for the CAPRA&PDE4D5/7/9_ BCR high-risk group (cut-off >0.835) demonstrates that this group of patients (*n* = 42) are at a substantially elevated risk to experience adverse outcomes after primary intervention with an 81% risk of post-surgical AP and a 64.2% risk of being diagnosed with any pathological ISUP Gleason grade >2 (Table 7).

Aggregating all data presented here, we conclude that the CAPRA&PDE4D5/7/9_ BCR prostate cancer risk model is not only significantly associated with biological, longitudinal outcomes as published earlier by us, but also adds independent value to predict adverse pathology outcomes based on pre-surgical scoring. Further, in a clinically low–intermediate prostate cancer risk cohort the CAPRA&PDE4D5/7/9_ BCR risk model demonstrated higher prediction power of post-surgical adverse outcomes compared to the clinically used metrics PRIAS which select low-risk patients for inclusion into active surveillance. At the same time the clinical-genomic vs. the PRIAS model defines 22.2% (*n* = 22 vs. *n* = 18 low-risk patients) and 38.1% (*n* = 29 vs. *n* = 21 low-risk patients) more patients as low risk to experience any type of adverse outcome compared to PRIAS in the low–intermediate (*n* = 84) and the entire (*n* = 151) patient cohorts, respectively.

## 4. Discussion

Previously, the mortality rate of prostate cancer was investigated in different Gleason score groups after central review of the historic Gleason score to contemporary grading criteria. Based on this data of approx. 700 patients with longitudinal follow-up, no patient with RP Gleason score ≤6 (ISUP grade 1) died of prostate cancer. A mortality rate per 1000 person years of 2.1 was observed for those with Gleason scores 3 + 4 (ISUP grade 2) on final pathology assessment, while the risk of disease specific death increased 3-, 7-, and 19-fold with post-surgery Gleason scores 4 + 3, 8, and 9 (ISUP grades 3, 4 and 5), respectively [27]. It is of note that tumors with pathology Gleason score 3 + 4 and low percentage (<10%) of Gleason grade 4 behave more similarly to Gleason score ≤6 tumors compared to those cancers with >40% Gleason grade 4. Consequently, a favorable intermediate-risk group with predominantly Gleason grade 3 in biopsy and limited additional adverse features (e.g., number of tumor positive needle biopsy cores and percentage of tumor within the core) was defined as being equivalent in risk characteristics to low-risk cancer patients who are candidates for management by active surveillance as an alternative to active treatment [28].

Contemporary used inclusion metrics such as the PRIAS criteria for selecting patients defined as low risk of disease with adverse pathological features and risk of progression over time are rule-based and significantly depend on the exclusiveness of a tumor grade ISUP Gleason 1. It is, however, well known that the diagnostic value of ISUP Gleason grade 1 as determined on a needle biopsy sample is limited due to sampling errors. During systematic US guided biopsies a maximum of 1% of prostate tissue is sampled. This may lead to missing smaller Gleason grade 4 components which is one of the reasons why >30–40% of all ISUP grade 1 tumors on diagnostic biopsy are upgraded to ≥ISUP grade 2 after post-surgical pathology [8,29,30]. Furthermore, intra- and inter-observer variability in pathological assessment of the Gleason score may also lead to grade migration between biopsy and radical prostatectomy.

The CAPRA score was initially developed to predict the risk of disease relapse after primary intervention using regression modeling using diagnostic input variables such as pre-operative PSA, the primary/secondary biopsy Gleason grade, etc. [26]. Although the risk score was not developed to predict post-surgical up-grading, we found the CAPRA score significantly associated in uni-variable logistic regression analysis with Gleason score up-grading after surgery. However, the CAPRA score demonstrated a limited power to predict adverse pathology including characteristics other than Gleason score migration.

We have previously demonstrated in multiple studies that the expression of the 3′-5′-cAMP phosphodiesterase PDE4D7 transcript in primary prostate tumor tissue is inversely associated with an elevated risk to experience post-surgical disease progression as detected by PSA relapse. Higher levels PDE4D7 expression may be protective against the spread of prostate tumor in a primary disease setting and subsequent detection of residual tumor by PSA recurrence. The exact potential of PDE4D7 in the process of tumor progression is subject to ongoing research. Based on the previous research, we developed a prognostic risk model in combination with the clinical CAPRA score with the normalized expression score of PDE4D7 as well as two other PDE4D long transcripts, namely PDE4D5 and PDE4D9, to predict the post-surgical risk of prostate cancer recurrence in a pre- and post-surgical diagnostic setting [22,23,24]. Similarly, as to our findings to predict BCR as an outcome endpoint, we observed that the addition of the long PDE4D transcripts PDE4D5, PDE4D7, and PDE4D9 to the CAPRA score to predict post-surgical adverse pathology improved the AUROC by up to 8 units compared to the clinical CAPRA score alone. We have previously shown that the expression of PDE4D7 is inversely correlated to an increase in ISUP pathology Gleason grade [21]. The observations here support the view that next to PDE4D7 also the PDE4D5 and PDE4D9 transcripts are likely associated with Gleason grading. Nonetheless, it seems beneficial to combine the genomics information as calculated from the PDE4D transcript quantitation with the clinical risk score CAPRA to derive the highest benefit for the prediction of adverse outcome on pathology or post-surgical disease recurrence. 

The NPV’s for several adverse pathology endpoints of the CAPRA&PDE4D5/7/9_BCR model based on the 0.1 risk score cut-off were generally 5-10% higher compared to the AS including metric PRIAS (low risk) in the pre-selected low-to-intermediate (*n* = 84) patient cohort, while the NPV’s of both models were comparable for the entire cohort (*n* = 151). However, in both cohorts the CAPRA&PDE4D5/7/9_BCR model included more patients into the low-risk group compared to the PRIAS criteria. Given the limited overlap of men that were selected by both models for low risk to experience adverse pathology or recurrence indicates to some extent perpendicular selection mechanisms of low-risk patients by either of the two models.

Several multi-gene signatures been developed in the past to predict adverse pathology outcome after prostate surgery [31,32,33]. In addition, nomograms or regression models which combine clinical variables have been established for the same purpose [34,35,36]. Here, we present a model consisting of both a clinical risk metric and a few genes with added value to the clinical model to predict adverse pathology outcome after radical prostatectomy. Phosphodiesterases are under widespread investigation as drug targets for the treatment of amongst others tumor diseases [37]. The indication of an elevated risk to experience disease progression over time by the CAPRA&PDE4D5/7/9 score may support the development of novel therapies based on modulation of PDE activities.

## 5. Conclusions 

Here, we demonstrate that our previously reported CAPRA&PDE4D5/7/9_BCR prostate cancer prognostic risk model was also significantly associated with adverse pathological outcome after radical prostatectomy of patients with primary disease. This was still the case after adjusting the multivariable logistic regression model for the clinical CAPRA score. The association was significant in the entire patient cohort but also in a selected clinically low-to-intermediate-risk cohort where high-risk patients and patients with clinically advanced cancer stages were excluded. In comparison to the PRIAS criteria, which are clinically used to select low-risk patients for inclusion into active surveillance, the CAPRA&PDE4D5/7/9_BCR model was more strongly associated with post-surgical adverse pathological outcomes and classified significantly more patients as low risk in both tested patient cohorts. Consequently, the risk assessment by use of the clinical-genomics CAPRA&PDE4D5/7/9_BCR model might be useful to support the inclusion into active surveillance as an alternative treatment option for low-risk prostate cancer patients.

## 6. Limitations

The retrospective nature of this study provides a potential limitation in the interpretation of the results. Patient inclusion might have been biased by this study design. Furthermore, all patients were diagnosed and treated years ago. The definition of grade and stage of the disease were updated during this time frame which may give rise to some variability in adjusting previous diagnostic measures to contemporary scorings.

## Figures and Tables

**Figure 1 cancers-15-00262-f001:**
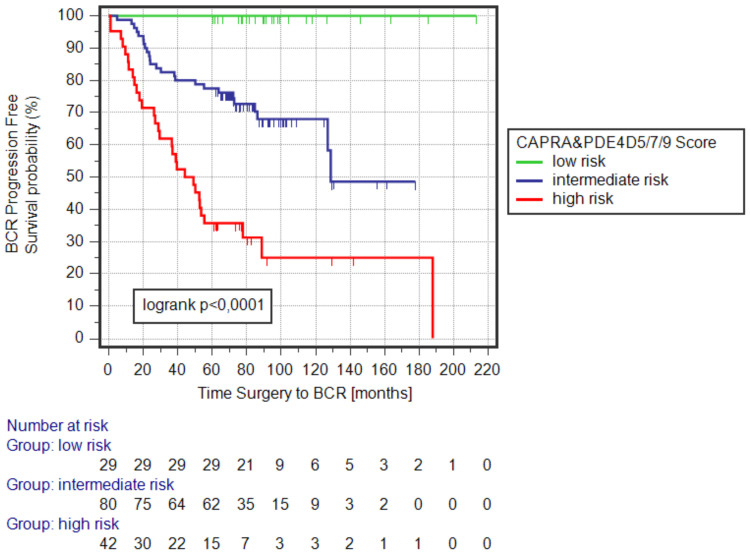
Kaplan–Meier survival analysis of the CAPRA&PDE4D5/7/9_ BCR regression model. For each patient we calculated the probability (p) to experience the endpoint of the regression model, namely post-surgical biochemical relapse (BCR) based on the individual’s pre-surgical CARPA score and the normalized expression values of the PDE4D transcripts PDE4D5, PDE4D7, and PDE4D9 as determined by RT-qPCR on diagnostic needle biopsy punch of the respective patient. The two cut-offs (low risk defined as CAPRA&PDE4D5/7/9_BCR scores <0.1; intermediate risk defined as CAPRA&PDE4D5/7/9_BCR scores 0.1 to 0.835; and high risk defined as CAPRA&PDE4D5/7/9_ BCR score >0.835) were selected such that low and intermediate-risk patient groups were exactly the same groups as previously published [24]. The here presented high-risk group represents the combined two highest risk groups as published before [24]. The number of patients in each sub-group is indicated for t = 0 (i.e., time of surgery) and every subsequent 20 months during the follow-up. End of follow-up is indicated by censoring. The statistical significance is given by the log rank *p*-value.

**Table 1 cancers-15-00262-t001:** The patient demographics for the entire patient cohort (*n* = 151), as well as the selected low-to-intermediate-risk cohort, and the low-risk patient group according to the CAPRA&PDE4D5/7/9_ BCR score is provided. IQR—interquartile range; RP—radical prostatectomy; ISUP grade—ISUP Gleason grade group; BCR—biochemical recurrence; SRT—salvage radiation therapy; SADT—salvage androgen deprivation therapy; PCSS—prostate cancer specific survival; OS—overall survival.

	Parameter	Entire Cohort (*n* = 151)	Low–Intermediate Risk Cohort (*n* = 84)	CAPRA&PDE4D579_low_risk (*n* = 22)
Demographic and Clinical Range (median; IQR)	Age range (at RP)	47.4–77.4 (64.9; 8.5)	52.3–76.9 (64.4; 8.3)	54.6–74.0 (64.6; 7.3)
	Preoperative PSA range	2.0–49.1 (8.1; 5.7)	2.0–49.1 (7.5; 4.9)	2.6–17.2 (5.7; 2.8)
	Prostate Volume range	13.6–148.0 (38.5; 19.2)	13.6–148.0 (39.5; 19.0)	15.4–105.7 (37.6; 18.5)
	PSA density range	0.03–1.6 (0.2; 0.17)	0.03–0.92 (0.2; 0.16)	0.03–0.48 (0.16; 0.12)
CAPRA Risk Category Number of Patients (%)	Low Risk (CARPA 0–2)	38 (25.2%)	32 (38.1%)	15 (68.2%)
	Intermediate Risk (CAPRA 3–5)	82(54.3%)	52 (61.9%)	7 (31.8%)
	High Risk (CAPRA > 5)	31 (20.5%)	0	0
Pre-Surgery Pathology Number of Patients (%)	Biopsy Gleason 3+3 (ISUP grade 1)	77 (51.0%)	53 (63.1%)	18
	Biopsy Gleason 3+4 (ISUP grade 2)	38 (25.2%)	23 (27.4%)	4
	Biopsy Gleason 4+3 (ISUP grade 3)	20 (13.2%)	4 (4.8%)	0
	Biopsy Gleason 8 (ISUP grade 4)	16 (10.6%)	4 (4.8%)	0
	Clinical Stage cT1	97 (64.2%)	84 (100%)	22 (100%)
	Clinical Stage cT2			
	Clinical Stage cT3	54 (35.8%)	0	0
Post-Surgery Pathology Number of Patients (%)	Pathology Gleason 3+3 (ISUP grade 1)	46 (30.5%)	34 (40.5%)	13 (59.1%)
	Pathology Gleason 3+4 (ISUP grade 2)	52 (34.4%)	32 (38.1%)	9 (40.9%)
	Pathology Gleason 4+3 (ISUP grade 3)	31 (20.5%)	11 (13.1%)	0
	Pathology Gleason 8 (ISUP grade 4)	22 (14.6%)	7 (8.3%)	0
	Pathology Stage pT2	88 (58.3%)	61 (72.6%)	17 (77.3%)
	Pathology Stage pT3	63 (41.7%)	22 (26.2%)	5 (22.7%)
	Pathology Stage pT4	0 (0%)	1 (1.2%)	0
	Positive Surgical Margins	33 (21.9%)	17 (20.2%)	4 (18.2%)
	Capsular Status penetrated with tumor cells	75/145 (51.7%)	23/82 (28.0%)	4/21 (19.0%)
	Positive Lymph Node Invasion	10 (6.6%)	2 (2.4%)	0
Follow-up (months)	Mean	73.7	87.0	102.8
	Median	73.6	82.1	92.8
BCR events (%)	BCR within 5 years	45 (29.8%)	16 (19.0%)	0
Salvage Treatment Events (%)	SRT within 5 years	12 (7.9%)	4 (4.8%)	0
	SADT within 5 years	16 (10.6%)	6 (7.1%)	0
Survival Events (%)	PCSS within 5 years	0 (0.7%)	0	0
	OS within 5 years	1 (0.7%)	0	0

**Table 2 cancers-15-00262-t002:** Testing of the mean difference CAPRA&PDE4D579_BCR score between adverse pathology outcomes. The Mann–Whitney test for independent samples was used to determine whether there is a difference in the mean risk score as calculated by the CAPRA&PDE4D579_BCR regression model between patient groups with difference post-surgical pathology outcomes. The tested outcomes were: (i) Adverse Pathology defined as any ISUP pathological Gleason grade >2, or pathological pT stage > pT3a, or tumor penetrated prostate capsular status, or pN1 disease; (ii) any ISUP pathological Gleason >2; (iii) any ISUP pathological Gleason >1. The number of patients per outcome group are indicated. The Mann–Whitney *p*-value is given.

Model	*n*	Outcome	*n* (Sample_1; Mean Probability p)	*n* (Sample_2; Mean Probability p)	*p*-Value
CAPRA&PDE4D5/7/9_BCR	84	Adverse Pathology (no vs. yes)	No (*n* = 52; 0.3)	Yes (*n* = 32; 0.5)	0.006
CAPRA&PDE4D5/7/9_BCR	84	RP ISUP Gleason (≤2 vs. ≥3)	≤2 (*n* = 66; 0.5)	≥3 (*n* = 18; 0.65)	<0.0001
CAPRA&PDE4D5/7/9_BCR	84	Pathology pT (≤pT3a vs. > pT3a)	≤pT3a (*n* = 76; 0.34)	>pT3a (*n* = 8; 0.71)	0.002

**Table 3 cancers-15-00262-t003:** Univariable logistic regression (UVLR) modeling of clinical and clinical-genomics risk models to post-surgical adverse pathology outcomes. The models tested were the clinical pre-surgical CAPRA score, the created logistic regression model combining the normalized PDE4D5, PDE4D7, and PDE4D9 scores (the PDE4D5/7/9_BCR model) to BCR as endpoint, the previously published CAPRA&PDE4D5/7/9_BCR risk model [24], and the active surveillance inclusion metric PRIAS [9]. The tested outcomes were: (i) Adverse Pathology defined as any ISUP pathological Gleason grade >2, or pathological pT stage > pT3a, or tumor penetrated prostate capsular status, or pN1 disease; (ii) any ISUP pathological Gleason >2; (iii) any ISUP pathological Gleason >1. OR–Odds ratio; OR (95% CI)–95% confidence interval of the Odds ratio; AUROC–Area Under the ROC Curve.

Model	*n*	Outcome	OR	OR (95% CI)	*p*-Value	AUROC
CAPRA Score	84	Adverse Pathology	1.4	0.95–2.1	0.09	0.6
PDE4D5/7/9_BCR			1.6	1.1–2.3	0.02	0.67
CAPRA&PDE4D579_BCR			1.4	1.1–1.7	0.01	0.68
CAPRA Score	84	RP ISUP Gleason >2	2.4	1.4–4.1	0.002	0.74
PDE4D5/7/9_BCR			2.5	1.5–4.1	0.0004	0.8
CAPRA&PDE4D579_BCR			1.9	1.4–2.7	0.0002	0.82
CAPRA Score	84	RP ISUP Gleason >1	1.6	1.1–2.4	0.02	0.65
PDE4D5/7/9_BCR			1.4	1.0–2.2	0.04	0.65
CAPRA&PDE4D579_BCR			1.4	1.1–1.7	0.01	0.68

**Table 4 cancers-15-00262-t004:** Multivariable logistic regression (MVLR) modeling of the PDE4D5/7/9_BCR risk model adjusted for the pre-surgical CAPRA score. The tested outcomes were: (i) Adverse Pathology defined as any ISUP pathological Gleason grade >2, or pathological pT stage > pT3a, or tumor penetrated prostate capsular status, or pN1 disease; (ii) any ISUP pathological Gleason >2; (iii) any ISUP pathological Gleason >1. OR–Odds ratio; OR (95% CI)–95% confidence interval of the Odds ratio; AUROC–Area Under the ROC Curve.

Model	*n*	Outcome	OR	OR (95% CI)	*p*-Value	AUROC
CAPRA Score	84	Adverse Pathology	1.3	0.8–1.9	0.3	0.67
PDE4D5/7/9_BCR			1.5	1.0–2.2	0.04	
CAPRA Score	84	RP ISUP Gleason >2	2	1.1–3.8	0.02	0.82
PDE4D5/7/9_BCR			2.2	1.3–3.7	0.004	
CAPRA Score	84	RP ISUP Gleason >1	1.5	0.99–2.3	0.06	0.68
PDE4D5/7/9_BCR			1.4	0.92–2.1	0.1	

**Table 5 cancers-15-00262-t005:** Negative Predictive Value (NPV) of the CAPRA&PDE4D5/7/9_BCR and the PRIAS active surveillance inclusion model (low-to-intermediate-risk cohort; *n* = 84). The cut-off 0.1 was selected for the CAPRA&PDE4D5/7/9_BCR regression model which separates the patients into a low-risk group (*n* = 22) vs. an intermediate–high-risk group (*n* = 62). The inclusion criteria for PRIAS selects 18 patients with low risk vs. 66 patients with intermediate–high risk. The tested endpoints are indicated. PSAD–PSA density (ng/ng/mL). Adverse Pathology is defined as any ISUP pathological Gleason grade >2, or pathological pT stage > pT3a, or tumor penetrated prostate capsular status, or pN1 disease. BCR–biochemical recurrence after surgery.

Adverse Pathology Outcome	NPV [%]	
Model	CAPRA&PDE4D5/7/9_BCR	PRIAS (Active Surveillance Inclusion)
Model cut-off	risk score < 0.1 (*n* = 22; 26.2%)	PSA < 10 ng/mL; PSAD < 0.2; ISUP Gleason grade 1; cT ≤ cT2; ≤2 tumor positive biopsy cores (*n* = 18; 21.4%)
Freedom of Adverse Pathology (AP)	82.0	72.1
ISUP pathology Gleason ≤2	100	94.4
ISUP pathology Gleason = 1	59.1	55.6
Pathological pT ≤3a	100	94.4
Pathological pT ≤2	77.3	77.8
Capsular Status (not penetrated)	81.8	76.5
Negative Surgical Margins	81.8	94.1
Freedom of Lymph Node Invasion	100	100
Freedom of BCR	100	94.4
Freedom of Secondary Therapy	100	100

**Table 6 cancers-15-00262-t006:** Negative Predictive Value (NPV) of the CAPRA&PDE4D5/7/9_BCR model (entire cohort; *n* = 151). The cut-off 0.1 was selected for the CAPRA&PDE4D5/7/9_BCR regression model which separates the patients into a low-risk group (*n* = 29) vs. an intermediate–high-risk group (*n* = 122). B) Positive Predictive Value (NPV) of the CAPRA&PDE4D5/7/9_BCR model (entire cohort; *n*=151). The cut-off 0.835 separates the patients into a low–intermediate risk group (*n* = 109) vs. a high-risk group (*n* = 42). The tested endpoints are indicated. PSAD–PSA density (ng/ng/mL). Adverse Pathology is defined as any ISUP pathological Gleason grade >2, or pathological pT stage > pT3a, or tumor penetrated prostate capsular status, or pN1 disease. BCR–biochemical recurrence after surgery.

Adverse Pathology Outcome	NPV [%]
Model	CAPRA&PDE4D5/7/9_BCR
Model cut-off	risk score <0.1 (*n* = 29; 19.2%)
Freedom of Adverse Pathology (AP)	72.4
ISUP pathology Gleason ≤2	93.1
ISUP pathology Gleason = 1	53.3
Pathological pT ≤3a	96.6
Pathological pT ≤2	75.9
Capsular Status (not penetrated)	75.9
Negative Surgical Margins	75.9
Freedom of Lymph Node Invasion	100
Freedom of BCR	100
Freedom of Secondary Therapy	100

**Table 7 cancers-15-00262-t007:** Positive Predictive Value (NPV) of the CAPRA&PDE4D5/7/9_BCR model (entire cohort; *n* = 151).

Adverse Pathology Outcome	PPV [%]
Model	CAPRA&PDE4D5/7/9_BCR
Model cut-off	risk score >0.835 (*n* = 42; 27.8%)
Adverse Pathology (AP)	81.0
ISUP pGleason >2	64.2
ISUP pGleason >1	92.9
Pathological pT >3a	38.1
Pathological pT >2	69,0
Capsular Status (penetrated)	58.5
Positive Surgical Margins	24.1
Lymph Node Invasion	14.3
BCR	71.4
Secondary Therapy	40.5

## Data Availability

In order to protect patient privacy, the clinical data of the individual patients have not been made available. The qPCR data generated to support the findings of this study may be released upon request to the corresponding author subject to respective scientific purpose of the use of the data and arrangement of respective data transfer agreements.

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
