# Peer review of "The CAPRA&PDE4D5/7/9 Prognostic Model Is Significantly Associated with Adverse Post-Surgical Pathology Outcomes"

_cancers, 2022, doi:10.3390/cancers15010262_

Round 1
Reviewer 1 Report
The paper entitled “The CAPRA&PDE4D5/7/9 Prognostic Model is Significantly Associated to Adverse Post-Surgical Pathology Outcomes” by Gulliver et al. is interesting. Biochemical recurrence (BCR) is the prelude of advanced metastatic/castration resistant prostate cancer and early detection would be valuable for early therapeutic intervention. The authors have developed a combined biomarker model for BCR including PDE4D gene isoform expression in the past few years. They retrospectively validated the prediction power of model for prostate cancer biochemical recurrence events in an independent patient cohort (n=151). The new model is significant in achieving high sensitivity and higher negative value.
Concerns
1. The PDE4D5/7/9 expression (score) are based on the highest grade biopsy punch from radical prostatectomy of patients. However, the majority of prostate cancer patients under surveillance may not have radical prostatectomy. This will limit very much the applicability of the assay for the majority of patients.
2. It would be informative to compare side by side with other BCR prediction models (e.g. combined serum biomarker quintet (Athanasiou et al. 2021), SCHLAP1(Kidd et al, 2021))
Minor concerns
Line 12: CAPRA & PDE4D5/7/9 ïƒ CAPRA&PDE4D5/7/9? this should be consistent thru the text, whichever is correct.
Line 16: n=151 ïƒ 151. n=84 ïƒ 84. There are others through the text.
Line 222-3: Figure 1. Kaplan-Meier survival analysis of the CAPRA&PDE4D5/7/9_ BCR regression model. Redundant sentences.
Line 313: Table 4. Multivariable logistic regression (MVLR) ïƒ MVLR
Line 512 and other references: the numbers of reference are duplicated.
Author Response
Please see attachment with our comments in red.

Reviewer 2 Report
In this study, The authors assessed whether the CAPRA+PDE4D5/7/9 score which was established as a prognostic model after prostatectomy could predict adverse pathological outcomes for low-/intermediate risk patients. They found that CAPRA + PDE4D5/7/9 score was significantly associated with adverse pathological outcomes and showed better predictive value than the CAPRA score or PDE4D5/7/9 score only. They showed that negative predictive value to speculate adverse pathology or biological recurrence was comparable between the CAPRA+PDE4D5/7/9 score and PRIAS inclusion criteria.
Comment)
Method)
1. (line 151) The formula and coefficient to calculate the CAPRA+PDE4D5/7/9 score should be shown because this is a key metric of this study.
2. In PRIAS criteria, which Gleason score criteria, 3+3 only before 2012 or 3+4 (<10%) after 2012, was used?
Result)
1. Figure 1 and the relevant data)
what is the difference between this figure and Figure 3 in ref # 24?
The author combined the risk score 3-4 and 4-5, but the other data seems to be exactly the same. what is new data provided?
2. (Table 2)
The caption was duplicated.
This table contains very little information and the number of adverse pathology patients seems wrong. The average CAPRA+PDE4D5/7/9 score of each group should be shown.
3. (Line 282) what is the meaning of "higher levels of significant association"? Is this mean the CAPRA+PDE4D5/7/9 model showed higher fitness? The author should measure the fitness of the model in this case.
4. (Line 288) The unit of AUROC is not "%".
5. (Line 292 and Line 422) The interpretation of this result should be done with caution. The PRIAS criterion is a just dichotomized variable and the purpose is to separate prostate cancer patients who are really safe for surveillance from the patients with a certain risk of progression. The authors should first dichotomize the CAPRA+PDE4D5/7/9 score with a certain cut-off (i.e. p <0.1) and compared the predictive value to PRIAS criteria.
6.(Table6) Patients who fulfill PRIAS criteria and CAPRA score >5 is unlikely. why the number of patients increased from 18 to 21?
7. (Line 415) It seems the PDE4D5/7/9 score is not the exception of sampling error. Is there any data that the PDE4D5/7/9 score represents the entire prostate cancer condition?
8. Because multiple predictive models or genomics have been published (BJU Int, 105 (2010), pp. 352-358, Cancer, 119 (2013), pp. 3992-4002, Eur Urol, 68 (2015), pp. 123-131, Urology, 166 (2022), pp 189-195 and so on.) what is the advantage of the CAPRA+PDE4D5/7/9 score?
Author Response
Please see the attachment with our comments in red

Round 2
Reviewer 2 Report
The author showed the appropriate revision.
Minor points to be revised)
1. Table 2 didn't reflect the revision, although the author said that they added the average logit value.
2. After the correction, the PRIAS column in table 5 and 6 were identical. I recommend omitting either one.
3. (line 286, "combination model demonstrated higher levels of significant associations") P-value (statistical significance) is not something to be compared, although the p-value in logistic regression partially reflects model fitness. A model with a higher association but a larger p-value is possible. I suggest changing the sentence to “the combination model demonstrated higher AUROC”.
Author Response
Please see the attachment and our responses in red font.
